# Binding of SARS-CoV-2 Structural Proteins to Hemoglobin and Myoglobin Studied by SPR and DR LPG

**DOI:** 10.3390/s23063346

**Published:** 2023-03-22

**Authors:** Georgi Dyankov, Petia Genova-Kalou, Tinko Eftimov, Sanaz Shoar Ghaffari, Vihar Mankov, Hristo Kisov, Petar Veselinov, Evdokia Hikova, Nikola Malinowski

**Affiliations:** 1Institute of Optical Materials and Technologies “Acad. J. Malinowski” (IOMT), Bulgarian Academy of Sciences (BAS), 109 “Acad. G. Bonchev” Str., 1113 Sofia, Bulgaria; 2Central Laboratory of Applied Physics, Bulgarian Academy of Sciences, 61 Sanct Peterburg Blvd., 4000 Plovdiv, Bulgaria; 3National Center of Infectious and Parasitic Diseases, 44A “Gen. Stoletov” Blvd., 1233 Sofia, Bulgaria; 4Photonics Research Center, Université du Québec en Outaouais, 101 Rue St-Jean Bosco, Gatineau, QC J8X 3G5, Canada

**Keywords:** biosensors, SARS-CoV-2, surface plasmon resonance, long period grating, structural proteins

## Abstract

One of the first clinical observations related to COVID-19 identified hematological dysfunctions. These were explained by theoretical modeling, which predicted that motifs from SARS-CoV-2 structural proteins could bind to porphyrin. At present, there is very little experimental data that could provide reliable information about possible interactions. The surface plasmon resonance (SPR) method and double resonance long period grating (DR LPG) were used to identify the binding of S/N protein and the receptor bind domain (RBD) to hemoglobin (Hb) and myoglobin (Mb). SPR transducers were functionalized with Hb and Mb, while LPG transducers, were only with Hb. Ligands were deposited by the matrix-assisted laser evaporation (MAPLE) method, which guarantees maximum interaction specificity. The experiments carried out showed S/N protein binding to Hb and Mb and RBD binding to Hb. Apart from that, they demonstrated that chemically-inactivated virus-like particles (VLPs) interact with Hb. The binding activity of S/N- and RBD proteins was assessed. It was found that protein binding fully inhibited heme functionality. The registered N protein binding to Hb/Mb is the first experimental fact that supports theoretical predictions. This fact suggests another function of this protein, not only binding RNA. The lower RBD binding activity reveals that other functional groups of S protein participate in the interaction. The high-affinity binding of these proteins to Hb provides an excellent opportunity for assessing the effectiveness of inhibitors targeting S/N proteins.

## 1. Introduction

Coronavirus disease-19 (COVID-19) is regarded as an infective-inflammatory disease that affects mainly the lungs. It is currently unclear whether the initial pathological viral process begins in the lungs, causing general hypoxia and iron dysmetabolism.

Hemoglobin/iron dysmetabolism is likely to be the leading process causing multiorgan disease and hypoxia. In this regard, it should be noted that there is a similarity between the distal amino acid sequence of the SARS-CoV-2 spike glycoprotein cytoplasmic tail and the hepcidin protein [1]. In this way, iron metabolism is changed, which leads to iron dysmetabolism and ferroptosis [2]. Apart from that, there are additional pathological metabolic pathways resulting from hemoglobin denaturation, based on predictions suggesting that the virus attacks the hemoglobin in red blood cells [3,4]. The SARS-CoV-2 virus interacts with the erythrocyte receptors such as cyclophilins, furins, TMPRSS2, and CD147 (in addition to the recognized ACE2) [5,6], penetrates the cells, and interacts with the hemoglobin.

Theoretical modeling has shown [7] that ORF1ab, ORF10, and ORF3a proteins could dissociate iron to form porphyrin while ORF8 and surface glycoprotein could bind to the porphyrin. The theoretical model used has shown that structural proteins do not bind to hemoglobin. However, it has been recently proved experimentally that S binds to Hb [8]. It has been found what part of RDB and the N-terminal domain (NTD) of the SARS-CoV-2 spike is responsible for binding the α and β chain of Hb. The assay involved SARS-CoV-2-infected Vero cell extracts incubated with Hb by means of the shotgun proteomics method. In silico analysis has been used to identify Hb binding motifs in the nucleoprotein.

Hemoglobin/iron dysmetabolism has been considered a leading process caused by a virus infection that results in multi-organ disease, so antibodies, vaccines, and inhibitors targeting potential pathophysiological pathways have been proposed [9]. Developing methods to study the possible molecular interactions is of primary importance. [10] explores the idea of a link between SARS-CoV-2 and protoporphyrin, so detection of the latter has been proposed as a diagnostic tool for infection.

Different sensing platforms can be used to study the binding behavior of SARS-CoV-2 proteins, two of the most effective being those based on surface plasmon resonance (SPR) and double resonance long-period gratings (DR LPG), both of which are highly sensitive to changes in the surrounding refractive index. Thus SPR has been successfully used for SARS-CoV-2 diagnosis [11,12,13]. Hemoglobin-functionalized SPR and DR LPGs transducers have been shown to be a sensitive platform for glucose sensing [14]. LPGs have been successfully used to detect SARS-CoV-2 S-proteins [15] and have been successfully applied for both bacteria and virus detection [16,17].

The aim of our study was to provide new additional experimental data on the mechanisms of binding of SARS-CoV-2 S/N structural proteins and RBD protein, as well as of VLPs to Hb/Mb. The aim of this study was also to predict or shed light upon the biological role of proteins and the clinical manifestation of viral disease, as well as on the possibilities of early diagnosis of infection. As a consequence, we formulated two hypotheses of biological and medical importance.

Given the availability of very few experimental results in the research area, our aim was, by using physical methods of detection, to confirm the results that have been previously obtained by biochemical methods. Unlike the experiments known so far in the field, ours were based on registering physical phenomena based on the direct molecular interactions of S/N proteins, RBD proteins, and VLPs with Hb/Mb, which take place on SPR and DR LPG transducers. Achieving corroborating results would remove any doubt regarding the correctness of such complex experiments.

Our study also aimed to establish a single standard protocol regarding ligands immobilization, bioactive substance manipulation, and measurement procedures. We evaluated the effectiveness of the protocols by obtaining consistent experimental results from two different detection systems, namely the SPR and the DR LPG systems.

## 2. Materials and Methods

### 2.1. Reagents and Materials

All the chemicals and reagents used were of analytical grade. The following SARS-CoV-2 specific structural proteins were used to evaluate the bimolecular interaction:

#### 2.1.1. SARS-CoV-2 Spike S1 Subunit Protein

The SARS-CoV-2 Spike S1 subunit protein fused to a C-terminal poly-histidine (6 × Histidine) tag with a tri-amino acid linker (Molecular weight (Mw)~123 kDa) was purchased from InvivoGen Company, (San Diego, CA, USA), 5, rue Jean Rodier F-31400 Toulouse France. Stock solutions for the experiments were prepared at an initial concentration of 100 µg/mL deionized water. Aliquots were prepared and stored at −20 °C until use. Working concentrations were propagated in fresh deionized water in the concentration range of 20 fM–800 fM.

#### 2.1.2. SARS-CoV-2 Nucleocapsid Protein

The SARS-CoV-2 nucleocapsid protein fused to IgG1 Fc tag with a TEV (Tobacco etch virus) sequence linker (Mw~79 kDa) was purchased from InvivoGen Company, USA. Stock solutions were prepared at an initial concentration of 100 µg/mL in deionized water. Aliquots were stored at −20 °C until use. Working concentrations of the stock solution were dissolved in fresh deionized water in the concentration range of 30 fM–810 fM.

#### 2.1.3. Hemoglobin and Myoglobin

Hb (from bovine blood, H2500) and Mb (from equine skeletal muscle, M0630) were purchased from Sigma-Aldrich (Sofia, Bulgaria, FOT Ltd.-representative).

#### 2.1.4. SARS-CoV-2 Spike-RBD (Receptor Binding Domain)

The soluble SARS-CoV-2 Spike-RBD protein fused to a C-terminal poly-histidine (6 × Histidine) tag with a 3 amino acid linker (Spike-RBD-His) (Mw~30 kDa) was purchased from InvivoGen Company, San Diego, CA, USA. Stock solutions for the experiments were prepared at an initial concentration of 100 µg/mL of deionized water. Aliquots were prepared and stored at −20 °C until use. Working concentrations of the stock solution were dissolved in fresh deionized water in the concentration range of 20–800 fM.

#### 2.1.5. Chemically-Inactivated Virus-like Particles (VLPs)

Chemically-inactivated VLPs, 500 µL tube, were purchased from (Microbiologics, San Diego, CA, USA). β-propiolactone chemical inactivator is used for SARS-CoV-2 virus inactivation. Working concentrations of the stock solution were dissolved in DEPC-treated water in the concentration range of 1:1000, 1:2500, 1:5000, and 1:10,000, which correspond to RNA copies/mL, respectively 411,800, 356,800, 42,640, and 44,530. Viral RNA was extracted by an automatic extraction system using the ExiPrep Dx Viral DNA/RNA kit (Bioneer, Daejeon, Republic of Korea) in accordance with the manufacturer’s instructions. The detection was provided by real-time RT-PCR using a CFX96 thermal cycler (Bio-Rad Laboratories, Inc., Hercules, CA, USA) and TaqPath COVID-19 CE-IVD PCR Kit (Thermo Fisher Scientific, Waltham, MA, USA).

### 2.2. Surface Plasmon Resonance Method

In our experiment SPR was excited on the surface of a gilded diffraction grating, as shown schematically in Figure 1. The gratings were supplied by DEMAX Ltd., Sofia, Bulgaria; for the purposes of the experiment, we covered them with 110 nm ± 10 nm gold film coating obtained by vacuum evaporation. SPR conditions were fulfilled for a P-polarized light beam that illuminated the grating at an incidence angle of 35 degrees. Typically, the resonance was excited in the range of 690–705 nm for a bare grating having 80 nm-high grooves and a period of 1.55 µm. The SPR chip represents the gilded grating covered with Hb or Mb of a certain thickness, as shown in Figure 1.

Details about the SPR measurement system based on spectrum readout can be found in [18]. The spectrometer accuracy measurement is 0.2 nm, while the overall accuracy of the resonance shift measurement of the SPR system is appraised at about 2 nm. The limit of detection (LOD) was evaluated by considering the accuracy of the spectrometer as well as the accuracy of the goniometer for setting up the angle of light incidence.

### 2.3. Double Resonance Long-Period Gratings (DR LPGs)

#### 2.3.1. Principle of Operation of Double Resonance Gratings around the Turning Point

A long-period grating (LPG) is a structure consisting of a sequence of modulations with a period Λ of the refractive index and/or the geometric dimensions of the core and cladding of a stripped optical fiber. The modulations cause coupling between the fundamental LP_01_ core mode of an effective refractive index *n*_01_ and an LP_0*p*_ higher-order cladding mode of the glass/air waveguide of an effective refractive index *n*_0*p*_. The higher order LP_0*p*_ cladding mode is attenuated in the subsequently coated fiber region, so a total loss is observed at the fiber end. The coupling reaches a maximum at a center wavelength
(1)$$\lambda_{c}=\Delta n_{eff}\Lambda \;\;\;\Delta n_{eff}=n_{01}-n_{0p}$$at which a minimum of the transmission spectrum is observed. As seen from (1), the center wavelength is proportional to the period which is a unique geometrical quantity, not dependent on the wavelength. Because of dispersion, for a given grating period Λ the effective refractive index difference Δ*n_eff_* can have two different values, Δ*n_eff_*_,1_ and Δ*n_eff_*_,2_, at two different wavelengths, due to which two resonance wavelengths *λ_c_*_1_ and *λ_c_*_2_ can occur for the same grating period, as shown in Figure 2a. Thus (1) becomes:(2)$$\lambda_{c1}=\Delta n_{eff,1}\Lambda \;\;\;\lambda_{c2}=\Delta n_{eff,2}\Lambda$$The point where *λ_c_*_1_ = *λ_c_*_2_ is indicated in Figure 2a as the turn-around point (TAP) wavelength at the grating period Λ_0*p*_ of the mode LP_0*p*_. For the 11th cladding mode, LP_0,11_ of the photosensitive fiber PS1250/1500 (Fibercore) Λ_0,11_ ≈ 207.7 μm. Figure 2b shows the evolution of the LPG spectrum with a period Λ at the turning point. The spectrum deepens and splits as either the temperature or the surrounding refractive index increase which leads to an increase in the spectral separation Δ*λ* = *λ_c_*_2_ − *λ_c_*_1_. The change of Δλ with SRI is the basis for label-free biosensors.

#### 2.3.2. Fabrication and Calibration Procedure

First, the DR LPGs were fabricated using a SYNRAD pulsed CO_2_ laser (*λ* = 10.6 μm, *f* = 20 kHz, at 13.1% to 14% of maximum power, a focal length of the scanning head is 10 cm and a spot size of about 100 μm). A NKT Photonics Super K Compact white light source (purchased from NKT Photonics Inc, 23 Drydock Ave, Boston, MA 02210, USA) and a Yokogawa AQ 6370C optical spectrum analyzer (OSA) were used to monitor the formation of the grating spectrum during the inscription process of the grating with Λ = 207.6 μm and *N* from 235 to 250. A section of a photosensitive PS 1250/1500 (Fibercore) with a length *l*_0_ = 10 cm was spliced between a lead-in and a lead-out SMF-28E single mode fiber each of 1 m. A weight of *m* = 4 g was attached to the fiber to guarantee repeatability in the fiber longitudinal strain.

Second, after fabrication, the LPG was immersed in water to check if the desired double resonance spectrum was achieved.

Third, the grating was placed in a 10% solution of HF acid and the split was reduced to the desired position around the turning point.

Fourth, after the grating was tuned closer to TAP, its sensitivity to SRI around the water was measured at a temperature of 23 °C. The spectral separation Δ*λ* = *λ_c_*_2_ − *λ_c_*_1_ between the minima as a function of the SRI *n* in the 1.33–1.35 range was found to be in the form
(3)$$\Delta \lambda =S_{n}n+A$$In (3) *A* is a constant and *S_n_* is the SRI sensitivity. Four of the fabricated DR LPGs under consecutive fabrication numbers P093, P095, P097, and P100 were further functionalized using the procedure described in the next Section 2.4. Table 1 below summarizes the individually measured SRI sensitivities *S_n_* and the type of ligand used to functionalize the gratings using a pulsed laser deposition technique.

After their functionalization, the gratings were placed in the measurement set-up, using the same weight and temperature as in the calibration measurement.

The spectral separation was measured in air and water immediately after immersion in water and 5 min later and is referred to as Δ*λ*_0_. The 5 min waiting period allows the grating to reach thermodynamic equilibrium in the liquid.

### 2.4. Deposition of Bioactive Ligands

Our experiment involved ligands of Hb and Mb molecules deposited on SPR gratings and only Hb deposited on LPG structures. Matrix-assisted laser evaporation (MAPLE) was used for the ligand deposition to avoid build-in matrix and related mediating molecules—the main causes for biosensor non-specific response. Details about the MAPLE technology and Hb/Mb immobilization can be found in [19,20]. Hemoglobin was deposited simultaneously on DR LPG structures (with their sensitivities measured in advance) and on SPR gratings. The LPG and SPR transducers were placed in a vacuum camera in groups of eight and were subsequently functionalized by the MAPLE technique. The SPR grating was Mb-functionalized at another MAPLE deposition.

For the direct Hb/Mb immobilization, a frozen solution target involving 5–7% Hb/Mb in deionized water was used. The main challenge in MAPLE deposition is whether the laser transfer preserves the bioactivity of the deposited molecules. To prove that the deposited protein layers preserve their functionality, the SPR biochips were blown with carbon monoxide (CO) and nitrogen monoxide (NO). The resonance spectral shift registered (in the range of 1–3 nm), which was due to the coupling of CO/NO with the heme/thiol groups, proved the bioactivity of the immobilized molecules.

### 2.5. Incubation of SPR Chips and Measurement Procedure

In addition to the simultaneous functionalization of the SPR gratings and the LPG structures, two other MAPLE depositions were performed to functionalize the SPR gratings. A total of 48 SPR biochips were examined in order to establish the measurement accuracy, out of which 36 were Hb-functionalized and 12 were Mb-functionalized.

The biochips functionalized with Hb were incubated as follows: 12—with S protein of different concentrations, 12—with N protein of different concentrations, and 12—with RBD protein of different concentrations.

The group of biochips functionalized with Mb was divided into two sets, and incubated with S- and N-proteins, respectively.

The biochips were incubated for 20 min at room temperature, then they were washed with deionized water (<2 µS/cm), after which the liquid phase was removed by centrifugation. All treatments were performed under the same conditions.

After the gilded diffraction gratings were functionalized, the plasmon resonances were measured at six different points on the biochip surface to evaluate the quality of the ligand layer. The spectral position of the resonances at each point was taken as a reference against which the shift due to the Hb/Mb—protein interaction was registered.

After incubation, the plasmon resonances were measured at the same si = x points on the biochip surface and the resonance wavelength shifts were estimated as differences from the reference resonances. Then the corresponding displacement average values and the absolute measurement errors were determined.

### 2.6. Incubation of LPG Structures and Measurement Procedure

DR LPG structures were incubated with SARS-CoV-2 structural proteins and RBD proteins at different concentrations. Four structures were incubated with S proteins, four—with N proteins, and four—with RBD proteins.

The spectra for each protein, starting from the lowest and going to the highest concentration, were consecutively measured. Measurements were performed immediately after the immersion into a particular concentration, 2.5 min later, and at the 5th min. The first LPG structure was incubated with all the protein concentrations. The incubation of the second structure started with the second higher concentration, then went up until the highest one was reached. The incubation of the third structure started with the third higher concentration. The fourth structure was incubated only with the highest concentration. The measurement results of each structure for the respective concentration were compared and good repeatability was observed. After measuring the highest concentration, the spectra in both water and air were re-measured.

Unlike SPR, DR LPG provides real-time measurement. To establish the correctness of the measurement procedure, the LPG transducer was washed after each incubation. The results were compared with those without washing.

The spectral separation Δ*λ* = *λ_c_*_2_ − *λ_c_*_1_ for each concentration was determined, after which the change of the spectral separation at a given concentration Δ*λ_i_* with respect to that in water Δ*λ*_0_ was defined as
(4)$$\delta \lambda_{i}=\Delta \lambda_{i}-\Delta \lambda_{0}$$

Since the accumulation of the detected protein increased the refractive index on the grating surface, the spectral changes *δλ_i_* were converted into refractive index changes *δn_i_* for the corresponding concentration by taking into account the sensitivity *S_n_* of the particular grating. From (4) and (3) it follows that
(5)$$\delta n_{i}=\frac{\delta \lambda_{i}}{S_{n}}$$Ultimately, the dependence *δn_i_*(*C_i_*) is plotted for each protein.

## 3. Results and Discussion

### 3.1. Results of the SPR Measurements

Figure 3a,b show the spectral shift of the plasmon resonance occurring at different concentrations, which results from the binding reaction between S/N proteins and hemoglobin and myoglobin, respectively. The binding affinity of the S proteins was higher than that of the N proteins for both the Hb and Mb ligands. This is entirely consistent with the finding [8] that the S protein has more molecular sequences that can interact with Hb, as compared to the N protein. To identify the binding of each specific functional group, the activity of the heme and thiol group was examined by blowing incubated biochips with CO and NO. No activity indicating an interaction with CO was detected, while the interaction with NO was identified by the same spectral shift as when using non-incubated biochips.

Apparently, binding to S/N proteins occurs in the heme group, inhibiting its functionality. The thiol group, responsible for NO metabolism, maintains its functionality. That being said, there must be another pathophysiological pathway reducing nitric oxide—a typical clinical syndrome of COVID-19 [21].

As expected, higher binding activity was observed in Hb than in Mb, since the number of heme groups in Hb is twice as large. It is worth mentioning the different trends in the concentration dependencies: Figure 3a shows a trend of continuous growth, which would be implausible, while Figure 3b demonstrates a saturation trend, as should be expected. Having in mind that the immobilized ligands have the same molar concentration, the thickness of the deposited Hb/Mb layers will be different. Therefore, the penetration of the plasmon field will be different, which accounts for the different sensitivities observed. Moreover, the established concentration dependencies were observed only in the indicated concentration range of the structural proteins and for the indicated thickness of the gold coating of the grating.

The registered N protein binding to Hb/Mb is the first experimental fact that supports theoretical predictions [8]. This fact suggests another function of this protein, not only binding RNA.RBD has been found to interact with Hb [8]. We also investigated this phenomenon. The SPR spectral shifts resulting from the binding of RBD to Hb are presented in Figure 4. Since the molar concentrations are similar, the RBD binding activity can be compared to that of the S protein (Figure 3a), since the biochips were elaborated simultaneously at one MAPLE deposition. The RBD binding activity is distinctly weaker.

The observed lower RBD binding activity suggests that binding to Hb is carried out not only by RBD and supports the conclusion that the N-terminal domain of the S protein is also involved [8]. The properties of N-/RBD proteins discussed above imply that the SARS-CoV-2 virus directly interacts with the heme group. To prove this assumption, chemically-inactivated VLPs were used—they are non-infectious particles composed of multiple proteins that typically form viral capsids mimicking the native viruses [22].

A pronounced Hb-VLP interaction was observed, illustrated by the SPR spectral shift shown in Figure 5 as a function of the VLP concentration in terms of RNA copies per mL.

It is worth noting that the interaction observed between Hb and VLPs at the lowest concentration provoked a response greatly exceeding LOD. Therefore, it might be concluded that the interaction with Hb might be able to diagnose symptomatic cases with lower viral loads. Unfortunately, this reaction is nonspecific and cannot be used for biosensor diagnosis involving clinical samples.

### 3.2. Results of DR LPG Measurements

The measurement procedure involved measuring the changes in the wavelength splits *δ*(*λ*) as a function of the concentration *C* of the N, S, and RBD proteins. Using (3) we determine the refractive index changes *δn* as:
(6)$$\delta n=\frac{\delta (\Delta \lambda )}{S_{n}}$$

Figure 6a presents the *δn*(*C*) dependencies for the N and S proteins while Figure 6b refers to the RBD proteins. Two subcases were considered. Measurements using LPG P100 differed from the others in that after each concentration measurement, the grating was kept in water for 5 min. The results are presented in solid squares and the refractive index changes are about twice lower compared to those observed following the standard procedure (solid rhombs). This suggests that water washes away the unbound proteins and does not change the trend of concentration dependence. Hence, washing always has to be performed.

Since the measurement conditions were not the same for S/N proteins and RBD interactions, they can hardly be compared. Regardless, the LPG results also demonstrated the higher binding activity of the S protein compared to that of the N protein (Figure 6a), similar to the one observed by SPR measurements (Figure 3a).

Using the Excel built-in fitting function, the *δn*(*C*) dependencies were fitted with a logarithmic function as
(7)$$\delta n(C)={\delta n}_{0}\ln C-{\delta N}_{0}={\delta n}_{0}\ln(\frac{C}{C_{0}})\;\mathrm{with}\;{\delta N}_{0}={\delta n}_{0}\ln C_{0}\;\mathrm{and}\;C_{0}=e^{\frac{{\delta N}_{0}}{{\delta n}_{0}}}$$
The fitting parameters *δn*_0_, *C*_0_, and *δN*_0_ from (7) and the coefficients of determination for each grating are given in comparison—Table 2.

## 4. Conclusions

The experimental results presented here imply that SARS-CoV-2 structural proteins and the virus itself actively bind here and completely inhibit its function. This causes poor clinical outcomes, as it is a ubiquitous molecule with an important role in many biological processes. Some of them, such as abnormal heme metabolism and oxygen transport can be registered long before the acute phase of infection. This is a consequence of the observed binding activity to Hb which is much higher than the binding to the monoclonal antibodies, as reported in a recent paper of ours [23]. In support of this hypothesis, it is worth mentioning that the common viral loads for COVID-19-positive patients have been found to be in the range of 10^6^–10^11^ RNA copies/mL [24], while we observed a pronounced binding at 10^4^ RNA copies/ml.

Due to their low specificity, the reactions studied cannot be used for diagnosing infection from clinical samples. However, their high affinity makes them perfect for assessing the effectiveness of inhibitors targeting S/N proteins. Since a number of potential antibodies directed to S-RBD, S-NTD, and N-NTD have been identified, such an evaluation approach can prevent further frustration in clinical trials. The real-time monitoring of biomolecular interactions performed by the DR LPG sensing platform is of great advantage in such studies.

The reported data demonstrate that SPR and DR LPG can detect proteins with a limit of detection reaching several tens of fM. This is in contrast to the 130 fM, achieved by the SPR detection based on antibody-antigen interaction, as reported in a recent paper of ours [23]. This comparison is admissible since constant concentrations of ligands and proteins were maintained in our experiments. The above-mentioned difference is due to the higher binding affinity of Hb/Mb. The detectable levels achieved at different sensing assays and the steady results obtained by the demonstrated biosensor platforms prove the correctness and consistency of the protocols applied for immobilization, bioactive material handling, and measurement procedures.

## Figures and Tables

**Figure 1 sensors-23-03346-f001:**
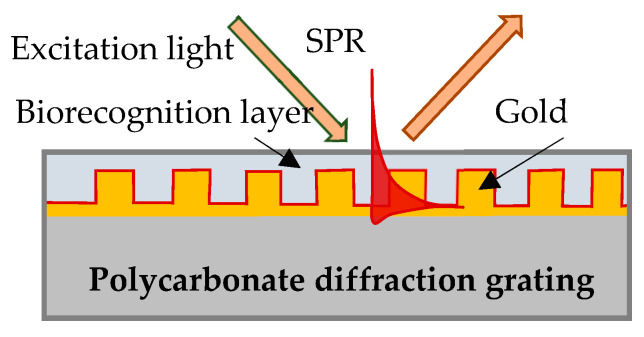
SPR chip: gilded diffraction grating with immobilized Hb/Mb.

**Figure 2 sensors-23-03346-f002:**
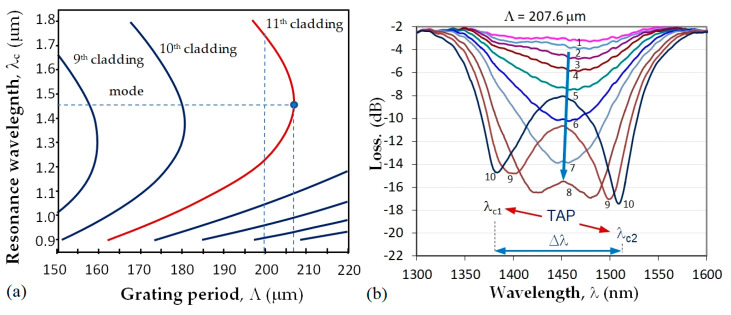
LLPG around turning point: (**a**) dispersion dependence and TAPs of photosensitive fiber PS1250/1500 (Fibercore) for different cladding modes; (**b**) TAP LPG splitting and transformation into a double resonance grating for the 11th cladding mode.

**Figure 3 sensors-23-03346-f003:**
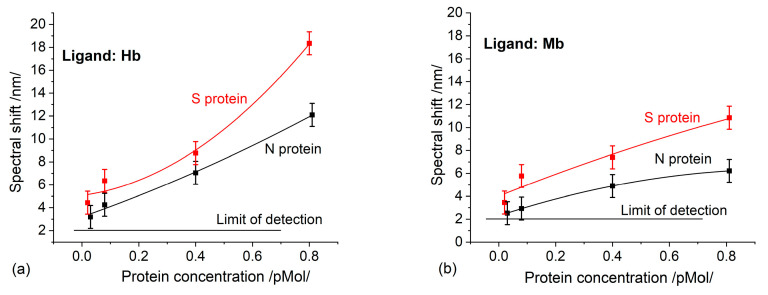
A spectral shift as a result of S/N proteins binding to: (**a**) Hb; (**b**) Mb.

**Figure 4 sensors-23-03346-f004:**
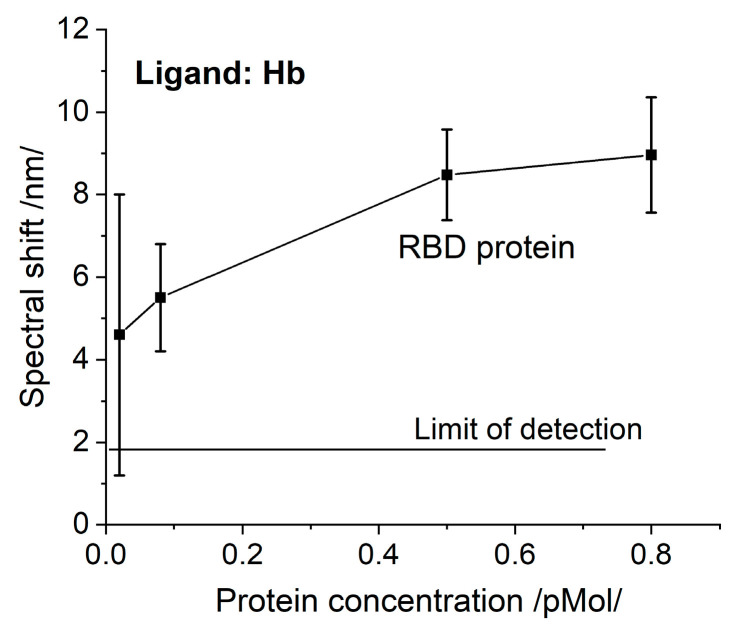
The spectral shift resulting from Hb/RBD interaction as a function of concentration.

**Figure 5 sensors-23-03346-f005:**
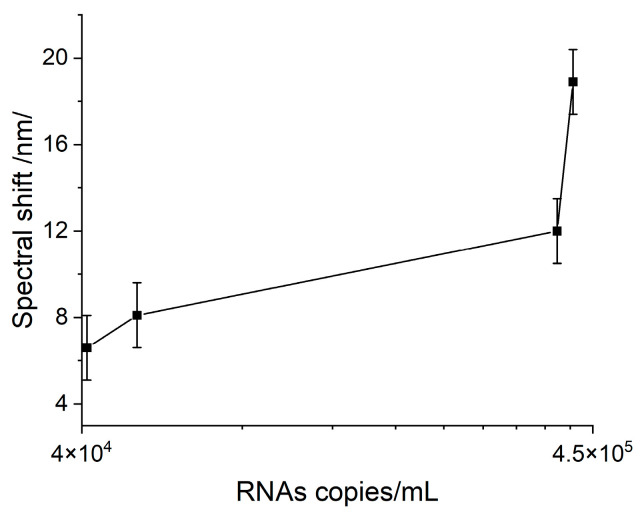
SPR response as a result of the Hb–VLP interaction at different concentrations.

**Figure 6 sensors-23-03346-f006:**
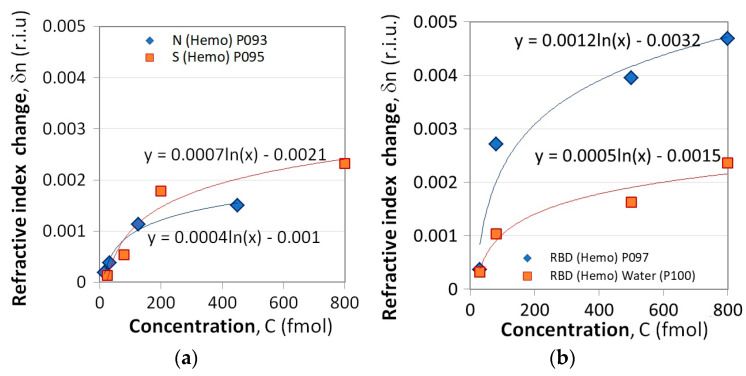
Refractive index change as a result of Hb binding to: (**a**) N and S proteins, (**b**) RBD proteins.

**Table 1 sensors-23-03346-t001:** Sensitivities *S_n_* to SRI, the type of functionalization, the detected proteins, and the maximum shifts of the DR LPGs used in the experiments.

DR LPG	P093	P095	P097	P100
Number of periods, N	250	235	240	240
SRI sensitivity *S_n_* (nm/r.i.u)	2661.1	2799.7	2025.3	2440.9
Functionalization	Hb	Hb	Hb	Hb
Protein	N	S	RBD	RBD & H_2_O
Maximum concentration (fmol)	450	800	800	800
Maximum shift (nm)	4	6.5	9.5	7.3

**Table 2 sensors-23-03346-t002:** Parameters for the logarithmic fit for the response to protein concentration.

DR LPG	P093	P095	P097	P100
Functionalization	Hb	Hb	Hb	Hb
*δn* _0_	0.0004	0.0007	0.0012	0.0007
*δN* _0_	0.0012	0.001	0.0032	0.0015
*C* _0_	20.086	4.173	14.392	8.524
*R^2^*	0.9829	0.9336	0.9273	0.9402

## Data Availability

Data and materials are available upon reasonable request to G. Dyankov (gdyankov@iomt.bas.bg).

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
