# Peer review of "Binding of SARS-CoV-2 Structural Proteins to Hemoglobin and Myoglobin Studied by SPR and DR LPG"

_sensors, 2023, doi:10.3390/s23063346_

Round 1
Reviewer 1 Report
The authors provide the S/N structural SARS-CoV-2 proteins and RBD protein bind heme. Although the manuscript described an interesting concept, it is premature with shortage of data to support the concept. In addition, the manuscript is not well prepared. Therefore, I do not recommend publication of the manuscript in the journal. Major concerns are as follows.
(1) Figure 1 was too vague. It is hard to understand the working mechanism from the present illustration.
(2) The novelty of this work was not clear. The advantages of this system (supported by direct experimental results) compared with the reported had not been addressed.
(3) Figure 3 is not sufficient to show the results of functional group combination, and more detailed data is needed.
(4) Figure 6 is not found in the manuscript.
(5) The format of the pictures in the manuscript should be consistent.
(6) It's hard to understand what the author is trying to say in the manuscript about the binding of SARS CoV-2 structural proteins to hemoglobin.
(7) I would suggest the authors to polish the manuscript carefully. There were many format errors over the main text.
Reviewer 2 Report
the article titled Binding of SARS CoV-2 structural proteins to hemoglobin and my- 2
oglobin studied by SPR and DR LPG) is accepted after consideration of the following comments.
!) abstract, page 1, line 29, authors should not use we I us etc.
2) page two . line 60, the abbreviations should be written as full name where firstly appear.
For example RDB and NTD.
3) page 2, line 64, the aim of this study is not clear please justify.
4) results and discussion should be merged in one section.
5) conclusion section should be added.
6) the rational of this study should be improved.
7) the literature survey about methods used should be mention.
8) references should be update as there are only three references 2021 and 2022.
Reviewer 3 Report
The authors studied the binding effect of SARS Cov-2 structural proteins to hemoglobin and myoglobin, and suggested the potential low-concentration detection of these proteins with the method of SPR and DR LPG. The experiments are designed with full scientific logic and the results are presented in an understandable way, thus we recommend that it be published after some minor improvements.
1. Can the authors provide more discussions on why this method is still promising despite it being non-specific? If the concentration of the target protein is low, wouldn't the detection be affected by other substances in the sample?
2. We seldom see abstracts written in a numbered order. It is suggested that the authors remove the numbering and write the abstract as a natural paragraph.
3. The authors should correct the spelling and case mistakes, such as "orf1ab" in line 56.
4. Figure 2 and Figure 5 do not have in-figure labels a) and b). Please also color the curves in Figure 2(b) by their order.
5. In Figure 3, how were the curves drawn? What kind of data fitting was applied here, and what is the physical meaning of the data fitting?
Round 2
Reviewer 1 Report
This work is well organized, conclusive, and apparently presented. Meanwhile, the author answered all the questions one by one and solved my previous confusion. The paper could be accepted for publication in Sensors.